# The Epidemiology of first and second-line drug-resistance *Mycobacterium tuberculosis* complex common species: Evidence from selected TB treatment initiating centers in Ethiopia

Biniyam Dagne[1,2]*, Kassu Desta[2], Rahel Fekade[1], Misikir Amare[1], Mengistu Tadesse[1], Getu Diriba[1], Betselot Zerihun[1], Melak Getu[1], Waganeh Sinshaw[1], Getachew Seid[1], Dinka Fekadu Gamtesa[1], Gebeyehu Assefa[3], Ayinalem Alemu[1]

1 Ethiopian Public Health Institute, Addis Ababa, Ethiopia, 2 Department of Medical Laboratory Science, College of Health Sciences, Addis Ababa University, Addis Ababa, Ethiopia, 3 Armauer Hansen Research Institute, Addis Ababa, Ethiopia

* biniyamd16@gmail.com

## Abstract

### Background

Drug-resistance in *Mycobacterium tuberculosis* complex remains a major health burden in human history and still is a major leading cause of death in developing countries including Ethiopia. Early detection of all forms of drug-resistant Tuberculosis(TB) is a key factor to reduce and contain the spread of these resistant strains.

### Methods

A health facility-based cross-sectional study was employed, based on demographic, clinical, and laboratory data collected from 204 patients with bacteriological confirmed TB. Sputum samples were analyzed using conventional TB culture and identification test followed by molecular species identification, and then phenotypic drug susceptibility tests. Data were entered using an excel spreadsheet and exported to SPSS version 20 for analysis. Descriptive analysis; frequencies, and proportions were computed.

### Results

Among the 204 sputum samples inoculated in culture media, *Mycobacterium* species were recovered from 165 specimens, with 160 *Mycobacterium tuberculosis* complex and five Non- Tuberculosis Mycobacterium(NTM) species. All *Mycobacterium tuberculosis* complex was found to be *M. tuberculosis*. Of the five NTM species, 2 *M.fortuitum*, 2 *M.intracellulare*, and 1 *M.gordonae* were identified. Among 160 species of *M. tuberculosis* isolates, 110 (68.8%) were resistant to any of the anti-TB drugs. The resistance pattern was; INH (109, 68.1%), RIF (99, 61.9%), STM (73,45.6%), and EMB (32,20.0%). Mono-resistance was found for INH (7,4.3%) and STM (1,0.6%). Ninety-nine (61.9%) isolates become MDR,

**Data Availability Statement:** All relevant data are within the manuscript and its Supporting Information files.

**Funding:** The author(s) received no specific funding for this work.

**Competing interests:** The authors have declared that no competing interests exist.

while resistance to any of the second-line anti-TB drugs was detected in 9 (5.6%) strains, with 8(5%) Pre-XDR and one (0.6%) XDR cases.

## Conclusion

Our findings highlight high frequencies of drug resistance to first and second-line anti-TB drugs.Determining the drug-resistance pattern of MTB is important for programmatic management of drug-resistant TB in Ethiopia. The circulating Pre-XDR and XDR case identified in the current study is alarming to the tuberculosis control program in the country.

## Introduction

The emergence of drug-resistant strains of *Mycobacterium tuberculosis*(MTB) is becoming a global challenge to tuberculosis (TB) control programs [1]. Drug-resistance TB (DR-TB) is caused by the *Mycobacterium tuberculosis* strain which is resistant to at least one anti-TB drug. Drug-resistance has mainly occurred as results of poor treatment outcomes, poor treatment adherence, poor quality of drugs, and poor infection control practices [2]. Nowadays, TB control is challenged by the emergence of drug-resistance to more anti-TB drugs resulting in multi-drug resistance (MDR), pre-extensively drug-resistant (Pre-XDR), and extensively drug-resistant (XDR) TB [2,3]. Mono resistance TB is caused by Mycobacterium tuberculosis strains that are resistance to at least one anti TB drugs. Poly resistance TB is caused by Mycobacterium tuberculosis strains that are resistance to at least two or more anti TB drugs other than rifampicin. Multidrug resistance (MDR) TB is caused by strains of *Mycobacterium tuberculosis (MTB)* that are resistant to at least the two potent anti-TB drugs; rifampicin and isoniazid. Pre-extensively drug-resistant TB is caused by *Mycobacterium tuberculosis* strains which are MDR plus resistant to any of the fluoroquinolones or one of the injectable second-line injectable anti-TB drugs. Extensively drug-resistant(XDR) TB is caused by *Mycobacterium tuberculosis* strains which are resistant to isoniazid and rifampin, plus any fluoroquinolone and at least one of three injectable second-line drugs (i.e., amikacin, kanamycin, or capreomycin) [1]. More recently, a more worrying situation has emerged with the description of *M. tuberculosis* strains that have been found resistant to all antibiotics that were available for testing, a situation labeled as totally drug-resistant (TDR)-TB [1,2].

In 2018, there were an estimated 484,000 incident cases of rifampicin resistance (RR)-TB and 377,520 cases of MDR-TB at the global level. Among the MDR/RR-TB cases, there were about214,000 deaths [2]. Despite the increase in testing, the number of MDR/RR-TB cases detected in 2018 only reached 38.6% (186,772) of the estimated one, and 13,068 patients with XDR-TB were reported worldwide [2,3]. Besides, on average, an estimated 8.5% of people with MDR-TB have XDR-TB [2]. Ethiopia is among the high TB, MDR-TB, and TB/HIV burden countries with an estimated 151 and 1.4 TB and MDR-TB cases per 100,000populations respectively. Based on the WHO 2019 global TB report, 0.71% of new TB cases and 16% of retreatment TB cases in Ethiopia were estimated to have MDR/RR-TB (WHO, 2019). In the same year, three XDR-TB cases were also reported [2–5].

The control of DR-TB, in any form, requires an accurate and prompt diagnosis of the type of resistance [6]. Universal access to drug susceptibility testing (DST) is one of the interventions recommended by WHO to reduce the burden, and for early initiation of the appropriate treatment [3,7]. The culture-based methods have been regarded as the gold standard for DST, and molecular methods provide rapid information on mutations in the *MTB* genome

associated with resistance to anti-TB drugs [4,8]. Currently, in Ethiopia there are more than 300 GneXpert facilities organized to detect MTB and rifampicin resistance.Throught the country there are 10 TB culture and DST laboratories including the National TB Reference Laboratory(NTRL). In the NTRL, there are different TB diagnostic/DST modalities including the phenotypic and molecular diagnostic methods(MGIT, GeneXpert, first-line and second-line LineProbe Assay). Determining the yield of DR-TB using phenotypic and genotypic DST methods are advantageous for scaling up efficient programmatic management and surveillance of DR-TB [9,10].

Even though Ethiopia is one of the high burden countries for DR-TB, there is no strong surveillance system monitoring the magnitude and the trend of the drug-resistance pattern of *Mycobacterium tuberculosis* across the country. The extent and the magnitude of the problem are not well studied. However, understanding the extent of DR-TB and the resistance pattern is important to design effective prevention and control strategies against its transmission and development of resistance to more anti-TB drugs. Thus, this study aimed to assess the drug-resistant pattern of *MTBC* for first-line and second-line anti-TB drugs among patients in selected treatment initiative centers (TICs) found in Ethiopia.

## Methods

### Study setting and study design

A health facility-based cross-sectional study was conducted among new and retreatment TB cases in eight TICs found in Ethiopia during the period between November 2019 and June 2020. In the country, there are about 53 TICs designed to manage the treatment of patients with drug-resistant TB. Eight TICs namely; St. Peter TB Specialized Hospital, *Shambu* hospital, *Gambella* Hospital, *Tulu Bolo* Hospital, *Ambo* Hospital, *Fitche* Hospital, *Asosa* Hospital, and *Yirgalem* Hospital were selected randomly in this study.

### Enrollment

All the consecutive adult pulmonary TB patients confirmed using any of the bacteriological diagnostic methods (Xpert MTB/RIF assay, smear microscopy) in the TICs during the study period that fulfilled the inclusion criteria were enrolled in the study.

**Inclusion criteria.** The inclusion criteria were; voluntary pulmonary TB patients with the age of ≥15 years, who were bacteriological confirmed new and re-treatment TB patients.

**Exclusion criteria.** Those patients with extrapulmonary TB, unable to produce sputum, and taking anti-TB treatment for more than one month were excluded.

### Sample size determination and sampling method

The required sample size was determined using a single population proportion by considering a 95% confidence level, 0.05 the level of significance, 5% marginal of error, and 14%prevalence [11]. By considering the estimated 10% non-response rate, the final sample size became 204. The consecutive sampling technique was applied until the achievement of the sample size within the given study period.

### Data collection procedure

A pre-tested data collection sheet was used to collect demographic and clinical profiles of the study participants. The data were collected by trained clinicians working in the TICs. Besides, the laboratory data on the drug-resistance pattern were captured using a laboratory logbook.

**Specimen collection.**  One 5–10 ml morning sputum sample was collected using a sterile sputum collection tube (falcon tube, 50 ml capacity) from each study participant under the instruction of laboratory professionals working in the TICs. The specimens were transported to National Tuberculosis Reference Laboratory, Ethiopian Public Health Institute for culture, and DST through a triple packaging sample transportation system.

## Laboratory analyses

**Specimen processing.**  All specimens were processed using the NALC-NaOH (N-acityl L-cystine sodium hydroxide sodium citrate) digestion decontamination technique described in the Global laboratory initiative (GLI)manual [12]. A final concentration of 1.5% NaOH was used for decontamination. After the centrifugation, the supernatant was decanted and the sediment was re-suspended in 2 ml of sterile phosphate buffer. Following processing, 0.5ml added to liquid media Mycobacteria Growyh indicater tube(MGIT) and 0.1ml inoculated to LJ(Lowenstein Jenson) media, AFB(Acid fast baccili) smear was prepared and stained with the Zehil Nelsson (ZN)Staining method. The tubes were incubated at 37˚Cfor up to 42 days for MGIT and 56 days for LJ media. The identification test was performed for all positive growths after confirming with smear microscopy. Brain heart infusion(BHI) agar plate was used to check for contamination. Finally, the SD bio line was used to differentiate Mycobacterium tuberculosis complex (MTBC) fromNonTuberculosisMycobacterium(NTM).

**Genotype Common Maycobacterium(CM), Aditional Species (AS), and *(Mycobacterium tuberculosis complex (MTBC)* assay Version 2.**  GenoType Mycobacterium CM and AS assey used as a species identification test for NTM and MTBC test Kit permits the identification of the different *Mycobacterium tuberculosis* complex species that were performed on clinical isolates and carried out according to the manufacturer instructions (Hain Life science GmbH, Nehren, Germany). Briefly, it was performed in four steps. DNA extraction (heat treatment and Genolyse kit), Polymerase Chain Reaction (PCR) reagent preparation, Multiplex amplification with biotinylated primers and Reverse hybridization (chemical denaturation of amplification products, hybridization of single-stranded, biotin-labeled amplicons to membrane-bound probes, stringent washing, the addition of streptavidin/alkaline phosphatase (AP) conjugate and staining mediated Substrate reaction). Evaluation and interpretation of results were evaluated by the presence of two control bands conjugate control and an amplification control. Positive results for all wild type probes of a gene and the absence of positive signal for mutation probes suggest strain sensitivity for the considered antibiotic. The absence of signal for at least one of the wild type probes with or without the presence of mutation probes hence indicates the resistance of tested strain to the considered antibiotic [7,12].

**MGIT 1st and 2nd line antibiotic susceptibility test.**  Susceptibility testing in the MGIT 960 system is based on the same principles as isolation from sputum (detection of growth). DST is performed using an AST (antibiotic susceptibility testing) set and TB exit, which consists of a Growth Control tube and one tube for each drug. A known concentration of a drug is added to the MGIT tube, along with the specimen, and growth is compared with a drug-free control of the same specimen. The critical concentrations of drugs for isoniazid (INH) 0.1µg/ml, rifampicin (RIF) 1.0µg/ml, streptomycin (STM) 2.0µg/ml, ethambutol (EMB) 5.0µg/ml; and second-line drugs: amikacin (AMK) 1.0µg/ml, kanamycin (KAN) 2.5µg/ml, capreomycin (CAP) 2.5µg/ml, ofloxacin (OFX) 2µg/ml, moxifloxacin (MOX) 0.25µg/ml and 1.0µg/ml. If the drug is active against the mycobacterial isolate (isolate susceptible), growth will be inhibited and fluorescence will be suppressed in the drug-containing tube; meanwhile, the drug-free control will grow and show increasing fluorescence. If the isolate is resistant, growth and its corresponding increase in fluorescence were evident in both the drug-containing and the

drug-free tube. The MGIT 960 system monitors these growth patterns and can automatically interpret results as susceptible or resistant. An isolate is defined as resistant if 1% or more of the test population grows in the presence of the critical concentration of the drug [13,14].

## Data quality control

Data collectors were trained to integrate the data collection sheet and the objective of the study. The data collection sheet was pre-tested on 5% of the sample size. Completeness and accuracy of the data were checked daily by the principal investigator. The sample was collected, stored, and transported based on the standard operating procedure of NTRL. The sterility of the culture media was checked by incubating the whole media at 37 0C for 48 hours and the performance of the media was cheeked by known susceptible *M. tuberculosis* (H37Rv). The sterility of sample processing reagents was cheeked by inoculating all reagents in a separate BHI. Positive and negative controls were included in every run of MTB culturing. For molecular operational activities sterile molecular grade water and reagent control was used as negative control and H37Rv ATCC 25177 used as a positive control. Moreover, to check the quality of the PCR and reverse hybridization process we see the presence of the amplification control (AC) band indicates that the DNA extraction and PCR procedures were carried out successfully and conjugate control (CC) documents two steps in the procedure. All laboratory results were recorded on a logbook during the study period. The collected data were analyzed and interpreted accordingly after it's checked for its completeness, accuracy, and clarity.

## Data analysis

Data were entered using an excel spreadsheet and exported to SPSS version 20 for analysis. Descriptive analysis; frequencies, proportions were used to explain the demographic, clinical profiles of the study participants, and the drug-resistance pattern to first-line and second-line anti-TB drugs. Finally, graphs and tables were used to describe the results.

## Ethical considerations

The study was approved by the Research and Ethics Review Committee of the Department of Medical Laboratory Sciences, College of Health Sciences; Addis Ababa University. Permission was obtained from the Ethiopian Public Health Institute and each participating health facilities. Informed consent was obtained from each study participant. All results were kept confidential; the participants were not identified by their name; instead, an appropriate coding system was used. The results were provided to the study participants and those in need of medical attention were communicated to respective physicians.

# Results

## Demographic and clinical characteristics

A total of 204 sputa were collected from new and retreatment cases attending eight treatment initiating centers in Ethiopia. Among the participants, 140 (68.6%) were males with a minimum and maximum age of 15 and 85 years respectively. The most frequent age-group was 25–34 years (83, 40.7%). Regarding the residence of the study participants, the majority (173, 84.8%) were urban dwellers. About 43.1% (88) were new cases, while the remaining were previously treated for TB. Based on the TB treatment classification, 88 (43.1%), 66(32.4%), 44 (21.6%), and 6(2.9%) cases were categorized under new, relapse, failure, and loss to follow-up cases respectively. Thirty-one (15.2%) of the participants were *HIV* Positive. Nearly half of the participants were smear-positive (104, 51%) **(Table 1).**

**Table 1. Demographic and clinical data of the study participants in selected drug-resistant TB Treatment Centers, Ethiopia, May 2020 (n = 204).**

| Variable | | Frequency | Proportion |
|---|---|---|---|
| **Gender** | | | |
| | Male | 140 | 68.6% |
| | Female | 64 | 31.4% |
| **Age** | | | |
| | 15–24 | 61 | 29.9% |
| | 25–34 | 83 | 40.7% |
| | 35–44 | 33 | 16.2% |
| | 45–54 | 14 | 6.8% |
| | $\geq$ 55 | 13 | 6.4% |
| **Residence** | | | |
| | Rural | 31 | 15.2% |
| | Urban | 173 | 84.8% |
| **Treatment History** | | | |
| | New | 88 | 43.1% |
| | First-line | 96 | 47.1% |
| | Second line | 20 | 9.8% |
| **TB Classification** | | | |
| | New | 88 | 43.1% |
| | Relapse | 66 | 32.4% |
| | Failure | 44 | 21.6% |
| | Defaulter | 6 | 2.9% |
| **HIV Status** | | | |
| | Negative | 173 | 84.8% |
| | Positive | 31 | 15.2% |
| **Smear Results** | | | |
| | Negative | 100 | 49.0% |
| | Positive (Scanty) | 23 | 11.3% |
| | Positive (+1) | 37 | 18.1% |
| | Positive (+2) | 21 | 10.3% |
| | Positive (+3) | 23 | 11.3% |
| **Living of Participant** | Addis Ababa | 86 | 42.2% |
| | Amhara | 13 | 6.4% |
| | BenishangulGumz | 5 | 2.4% |
| | Gambella | 12 | 5.9% |
| | Oromia | 35 | 17.1% |
| | SNNP | 53 | 26.0% |

## Culture and species identification

All 204 sputum samples were cultured using both solid media (LJ) and liquid media (MGIT). Using the comprehensive result of both culture media, 35 (16.9%) and 4 (1.9%) sputum samples gave culture-negative and contamination results respectively which are excluded from the final analysis. Thus, 165 (79.7%) sputum samples with culture-positive results are included in the final analysis. Through MPT64 antigen identification, 160 cases become MTBC, while 5 cases become NTM. Based on specious identification, all 160 MTBC cases become *Mycobacterium tuberculosis*, while among the five NTM cases two were *M.fortuitum*, two were *M.intracellulare*, and one was *M.gordonae* (**Table 2** and **Fig 1**).

**Table 2. Drug resistance pattern to first-line and second-line anti-TB drugs in selected Drug-resistant TB Treatment Centers, May 2020, Ethiopia (n = 160).**

| Resistance Pattern | New cases. frequency (%) | Retreatment cases. frequency (%) | All cases. frequency (%) |
|---|---|---|---|
| **Resistance Pattern** | | | |
| Total | N = 64 | N = 96 | 160 |
| Susceptible | 21 (32.8) | 29 (30.2) | 50 (31.2) |
| Any Resistance | 43 (67.2) | 67 (69.8) | 110 (68.8) |
| **All Resistance** | | | |
| RIF | 37 (57.8) | 62 (64.5) | 99 (61.9) |
| INH | 41 (64) | 68 (70.8) | 109 (68.1) |
| STM | 27 (42.2) | 46 (47.9) | 73 (45.6) |
| EMB | 12 (18.8) | 20 (20.8) | 32 (20.0) |
| **Mono Resistance** | | | |
| RIF | 0 | 0 | 0 |
| INH | 4 (6.2) | 3 (3.1) | 7 (4.3) |
| EMB | 0 | 0 | 0 |
| STM | 1 (1.5) | 0 | 1 (0.6) |
| **All MDR** | 37 (57.8) | 62 (64.5) | 99 (61.9) |
| INH+RIF (Only) | 10 (15.6) | 15 (15.6) | 25 (15.6) |
| INH+RIF+STM | 15 (23.4) | 28 (29.1) | 43 (26.9) |
| INH+RIF+EMB+STM | 11(17.2) | 17 (17.7) | 28 (17.5) |
| INH+RIF+EMB | 1 (1.5) | 2 (2.1) | 3 (1.9) |
| **Poly Resistance** | | | |
| INH+STM | 1(1.5) | 1 (1) | 2 (1.3) |
| INH+EMB | 0 | 1 (1) | 1 (0.6) |
| **Pre XDR** | 1(1.5) | 7 (7.3) | 8 (5) |
| MDR+AMK | 0 | 0 | 0 |
| MDR+CAP | 0 | 1 (1) | 1 (0.6) |
| MDR+KAN | 0 | 1 (1) | 1 (0.6) |
| MDR+MOX | 0 | 0 | 0 |
| MDR+OFX | 1(1.5) | 2 (2.1) | 3 (1.8) |
| MDR+MOX+OFX | 0 | 3 (3.1) | 3 (1.8) |
| **XDR** (MDR+CAP +OFX) | 0 | 1 (1) | 1(0.6) |

## Drug-resistance pattern to first-line anti-TB drugs

Drug susceptibility test was performed for 160 MTBC isolates for four first-line anti-TB drugs namely, Rifampicin (RIF), Isoniazid (INH), Ethambutol (EMB), and Streptomycin (STM). Resistance to any of the first-line anti-TB drugs was detected in 68.8% (110) of the isolates. The resistance pattern for four first-line anti-TB drugs was; INH (109, 68.1%), RIF (99, 61.9%), STM (73, 45.6%), and EMB (32, 20.0%). Mono-resistance was found for INH (7, 4.3%) and STM (1, 0.6%). Multi-drug resistance (MDR) was detected in 61.9% (99) of the MTB isolates. Regarding the combination of resistance to the drugs, 25 (15.6%) isolates were resistant to RIF and INH only, while 43(26.9%) isolates become resistant to INH, RIF, and STM. Besides, three (1.9%) isolates become resistant to INH, RIF, and EMB. Among all 160 MTBC cases, 28 (17.5%) strains become resistant to all the four first-line anti-TB drugs (INH+RIF+STM +EMB). Poly resistance was detected from three strains (1.9%); INH+STM (2, 1.3%), and INH +EMB (1,0.6%) (**Table 2**).

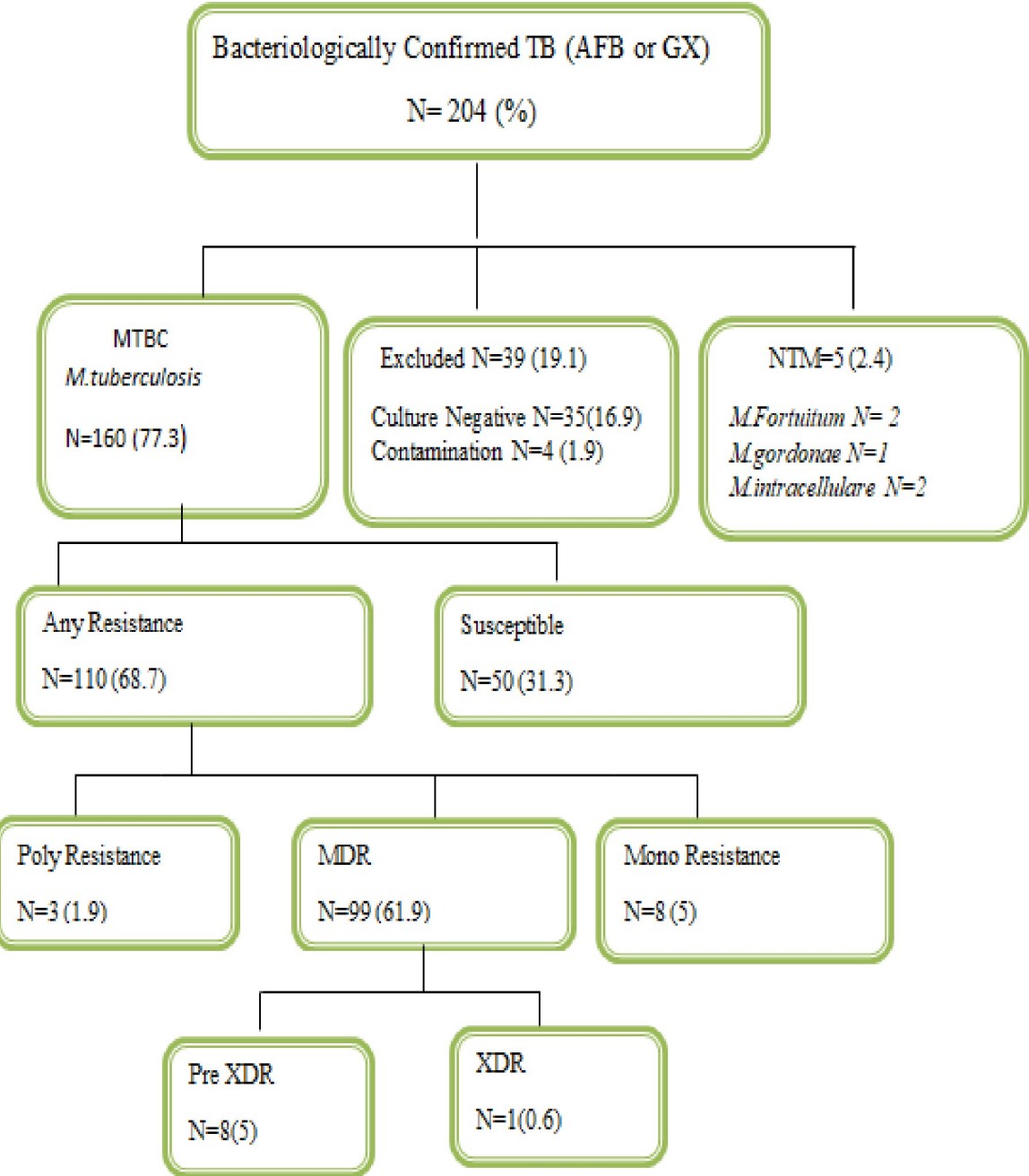

**Fig 1. Flowchart followed to assess the drug-resistance pattern of *Mycobacterium tuberculosis*.**

### Drug-resistance pattern to second-line anti-TB drugs

A second-line drug susceptibility test was performed for all MDR cases (99, 61.9%). Resistance profile was determined for Fluoroquinolones (FLQs) such as Moxifloxacin (MOX), Ofloxacin (OFX), and second-line injectable drugs (SLIDs) such as Amikacin (AMK), Capreomycin (CAP), and Kanamycin (KAN). Resistance to any of the second-line anti-TB drugs was detected in 9 (5.6%) strains. Eight cases (5%) were pre-XDR, while one case (0.6%) was XDR. Among the pre-XDR cases, six (3.8%) cases were resistant to any of the FLQ, while two (1.3%)

cases were resistant to any of the SLIDs. All six FLQ resistant strains become resistant to OFX. Besides the XDR case was also resistant to OFX. Any resistance to MOX was detected in three strains which are also resistant to OFX. Any resistance to SLIDs was detected in three strains (1.3%), 2 were pre-XDR and one was XDR. Resistance to Capreomycin was detected in two cases, while KAN resistance was detected in a single case. However, AMK resistance was not detected. The resistance pattern for the single XDR-TB case was; INH+RIF+STM+EMB+CAP +OFX (**Table 2** and **Fig 2**).

### Drug-resistance pattern based on treatment history

Among the valid 160 DST results, 64(40%) were new cases and 96 (60%) were retreatment cases. Of the new cases, 43 (67.2%) become resistant to any of the first-line drugs and 1(1.5%) were resistant to any of the second-line drugs. Among the retreatment cases, 67 (69.8%) become resistant to any of the first-line drugs and 8 (8.3%) were resistant to any of the second-line drugs. Among the retreatment cases who took first-line anti-TB drugs, 67/96 (69.8%)

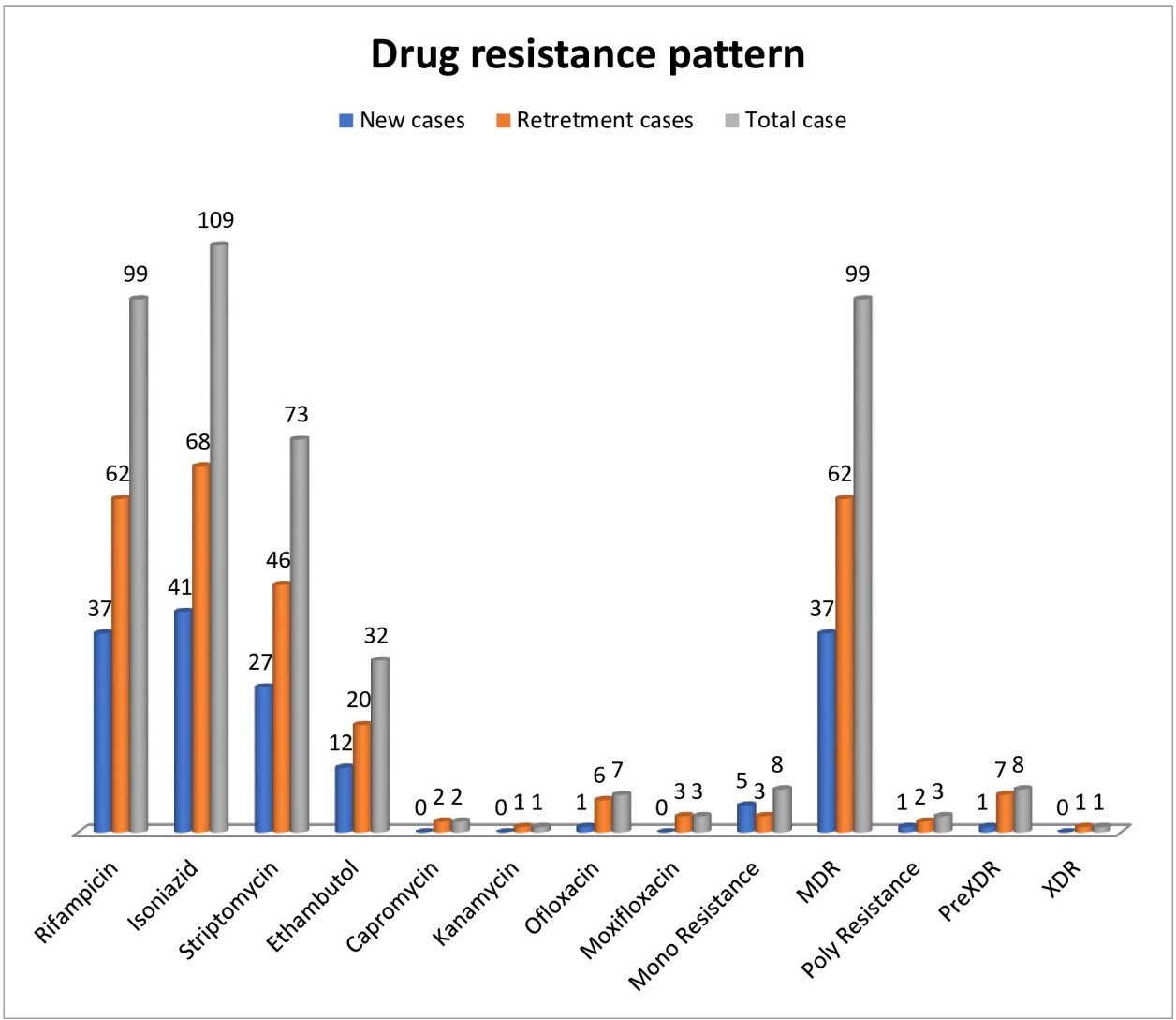

**Fig 2. Frequency of drug-resistance pattern for first-line and second-line anti-TB Drugs.**

become resistant to any of the first-line drugs and 4/96 (4.2%) were resistant to any of the second-line drugs. Among those who took second-line anti-TB drugs,4/20 (20%) become resistant to any of the second-line drugs (**Table 2** and **Fig 2**).

## Discussion

Drug-resistance strains of MTB are transmitted in the community, replacing susceptible strains and consequently making first-line regimens inadequate for achieving high cure rates. The efficiency of the current TB control program in any country is assayed by a drug-resistant pattern. The spread of drug-resistant tuberculosis can only be prevented by the rapid identification of these cases and treatment with a combination of effective drugs. Meanwhile, in the present study, we analyzed resistance to nine anti TB drugs (RIF, INH, STM, EMB, MOX, OFX, AMK, CAP, and KAN) detected in 160 isolates, using the golden standard MGIT phenotypic method. This study revealed that 68.8% (110), 61.9% (99), 5% (8), and 0.6% (1) of MTB strains became resistant to any of the anti-TB drugs, MDR, pre-XDR, and XDR respectively.

In the present study, all culture-positive MTBC isolates were identified as *M.tuberculosis*- whereas five NTM isolates were identified as *M.gordonae*, *M.fortuitum*, and *M.intracellulare.* This study coincides with a previous study conducted in Southern Ethiopia and Bangladesh [15,16]. This result was dissimilar with the studies conducted in North West Ethiopia, Turkey, Djibouti, and Ghana (97.6% *M.tuberculosis and* 2.4% *M.africanum*) [17–20]. The difference among species between different study settings may be due to differences in diagnostic methods, geographic area, and study participants.

In this study, the demographic data of participants from drug-resistance TB treatment centers indicated that bacteriologically confirmed cases were higher in previously treated cases 96 (60%) than new cases 64(40%) and all so this study showed a high rate of drug-resistance, 69.8% among previously treated cases than new cases as determined with a phenotypic method. The drug-resistance pattern to at least one of the anti-TB drugs (110, 68.8%) among the newly diagnosed TB cases 43/160 (26.8%), and from previously treated TB cases (67/160 (37.5%) identified in this study is consistent with previous study conducted in Addis Ababa (resistance to any drug, 54%) [21]. Similarly, a report by Nafees et al., from Pakistan, reported 62.6% of any drug-resistance [22]. Some countries like Germany and the US have very low drug-resistance cases with 12.7% and 1% respectively [23,24]. The results of our study showed that the highest proportion of drug-resistance identified for INH (68.1%), followed by RIF (61.9%) cases. Resistance to STM was found in (45.6%) and to EMB (20%). Comparable proportions were reported in other areas of Ethiopia resistance to INH was (51.4%), RIF (32.9%), STM (42.9%), and EMB (28.6%) cases of tuberculosis [25]. However, lower levels of resistance reported in Iran to INH (10%), RIF (11.8%), STM (10.7%), and EMB (3.2%) cases of tuberculosis [26]. The higher proportion of drug-resistance in the current study might be due to the nature of the study participant because the participants in our study were from drug-resistance TB treatment initiating centers. Besides, the present study indicated that a high rate (68.1%) of INH resistance is alarming since it is a potent first-line drug used though out the course of treatment. INH is also used as chemoprophylaxis of TB especially in immune-compromised Patients, TB High-risk group. Population and it acts on the dormant stage of TB, therefore, used as monotherapy for treating latent TB infection.

Any resistance to Rifampicin was found in (61.9%) of cases in this study. RIF is also a front line anti TB drug. The reason for high resistance may be due to conditions that alter the metabolism of the drug, treatment interruption, and using in the treatment of other bacterial diseases. Any resistance to Streptomycin was found in (45.6%) of cases. STM is the first anti-tuberculosis drug. The reason for this high resistance might be due to previously used as

monotherapy for treating TB for a long period, using for treatment of endocarditis, Plague, and Brucellosis.

In this study, 99/160 (61.9%) were MDR cases. The emergence of drug-resistance is a great public health threat across the globe and access to universal DST is also another concern. A higher rate of MDR was also reported in Ethiopia (87%), Pakistan (75.7%), and India (69.4%) [27–29]. Lower rate reported in Central Ethiopia (39.4), Sudan (51.8%) and Germany (16.7%) [21,23,30]. The variation in proportion among different study settings could be due to the study population because hospital-based studies particularly in drug resistance TB referral centers increase the finding of MDR-TB cases.

In the present study out of 99 MDR-TB cases, 8(8.1%) and 1(1%) were found to be Pre-XDR and XDR cases respectively. A similar rate of Pre-XDR was reported from Northern Ethiopia (5.7%) and Brazil (9.19%) [27,31]. A higher rate was reported from China (30.9%) and Zimbabwe (27%) [32,33]. The difference in finding may be due to the study groups because most patients in the TICs were presumptive for drug-resistance TB, geographic differences, and different levels of the health care delivery system in various countries.

The study had certain limitations. First, the study participants were all patients visiting selected drug-resistance TB treatment Center in Ethiopia. Findings from such a selected population may not indicate the true burden of the problem at the community level. The sample size was small relative which might be difficult to estimate the true proportion of drug-resistance to second-line anti-TB drugs.

## Conclusion

The findings of this study on the DST pattern showed that the magnitude of drug-resistant TB in the previously treated TB cases was higher as compared to new cases. The highest proportion of drug resistance was detected for INH followed by RIF, EMB, and STM. Furthermore, for all RIF and MDR TB cases, second-line DST was also conducted and the finding indicated that a higher frequency of OFX resistance identified followed by MXF, few CAP, KAN, and No AMK resistance cases detected. Finally, eight Pre-XDR cases and one XDR Cases were identified. Circulating Pre-XDR cases and XDR case is alarming to the tuberculosis control program in the country.

## Supporting information

**S1 File.**
(SAV)

## Acknowledgments

We would like to acknowledge Addis Ababa University, Department of Medical Laboratory Sciences, the staff of National Tuberculosis Reference Laboratory, Ethiopian Public Health Institute, and the staff and administrators of each participating health facilities. Finally, this study was conducted through the willingness of the study participants.

## Author Contributions

**Conceptualization:** Biniyam Dagne, Kassu Desta.

**Data curation:** Biniyam Dagne, Kassu Desta, Ayinalem Alemu.

**Formal analysis:** Biniyam Dagne, Kassu Desta, Rahel Fekade.

**Investigation:** Biniyam Dagne, Rahel Fekade, Misikir Amare, Mengistu Tadesse, Getu Diriba, Betselot Zerihun, Melak Getu, Waganeh Sinshaw, Getachew Seid, Dinka Fekadu Gamtesa, Gebeyehu Assefa.

**Methodology:** Biniyam Dagne, Kassu Desta.

**Supervision:** Kassu Desta.

**Writing – original draft:** Biniyam Dagne.

**Writing – review & editing:** Biniyam Dagne, Kassu Desta, Ayinalem Alemu.

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
