## [Decision Letter · Decision Letter 0]

9 Dec 2020

PONE-D-20-34803

The Epidemiology of first and second-line drug-resistance Mycobacterium tuberculosis complex common species: Evidence from selected TB treatment initiating centers in Ethiopia.

PLOS ONE

Dear Dr. Dagne,

Thank you for submitting your manuscript to PLOS ONE. After careful consideration, we feel that it has merit but does not fully meet PLOS ONE’s publication criteria as it currently stands. Therefore, we invite you to submit a revised version of the manuscript that addresses the points raised during the review process.

We look forward to receiving your revised manuscript.

Kind regards,

Seyed Ehtesham Hasnain

Academic Editor

PLOS ONE

Journal Requirements:

2. We noticed you have some minor occurrence of overlapping text with the following previous publications, which needs to be addressed:

https://synapse.koreamed.org/articles/1091312

https://www.scirp.org/pdf/AiM20120300014_34905242.pdf

http://etd.aau.edu.et/bitstream/handle/123456789/7091/17.GENENE%20BEKELE.pdf?sequence=1

https://www.mdpi.com/2079-6382/3/3/317/htm

In your revision ensure you cite all your sources (including your own works), and quote or rephrase any duplicated text outside the methods section. Further consideration is dependent on these concerns being addressed.

Additional Editor Comments:

Major Revision

Reviewers' comments:

Reviewer's Responses to Questions

**Comments to the Author**

1. Is the manuscript technically sound, and do the data support the conclusions?

Reviewer #1: Partly

Reviewer #2: Yes

Reviewer #3: Yes

2. Has the statistical analysis been performed appropriately and rigorously? 

Reviewer #1: I Don't Know

Reviewer #2: Yes

Reviewer #3: Yes

3. Have the authors made all data underlying the findings in their manuscript fully available?

Reviewer #1: Yes

Reviewer #2: Yes

Reviewer #3: Yes

4. Is the manuscript presented in an intelligible fashion and written in standard English?

Reviewer #1: Yes

Reviewer #2: Yes

Reviewer #3: Yes

5. Review Comments to the Author

Reviewer #1: Comments:

While reviewing this article, actual findings are quite common and well known facts for TB prevalent countries worldwide. However, the DR-TB facts may differ for Ethiopia. Data may provide the treatment utility and recommendations in present scenario for TB programme successfully progresses. Following are some suggestions which can make the manuscript stronger and beneficial for the readers:

Abstract:

• Word limits excided as per Journal guidelines. Should be modified.

• Conclusive statement should be modified to give actual interpretation of present study. Its known fact that the higher prevalence of DR TB always associated with previously treated TB patients.

Introduction:

• Add reference in line no. 67.

• Abbreviation must require for Mycobacterium tuberculosis (M. tuberculosis/ MTB) upon first appearance in the text (Line no 63).

• Author should advice to write few points about different molecular diagnostics modalities used presently in the referral laboratories.

• Abbreviation required for XDR-TB.

Method:

• While recruitment of subject, authors were enrollment only TB confirmed patients using Xpert MTB/RIF and Smear microscopy, then how they get NTM among 204 TB confirmed specimens (same statement written in abstract section)?

• Reference numbering required corrections and rearrangement.

• Abbreviation must require for GLI, LJ, ZN, MGIT, NTM, MTBC, NTRL, BHI, MTB, PCR….. upon first appearance in the text.

• While writing Genotype CM, AS and MTBC assays, author should advice to write it separately. Because CM/AS kit used for NTM speciation and MTBDR V.2 assay used for indentify DR-TB.

• In patient history, alcoholism and smoker status of patient as well as house hold contact should be noted if possible. Because these habits are strongly associated with TB disease and attributed to DRTB and recurrence.

Results:

• Authors were excluded all the contaminated culture from the present study. During NTM isolation M. fortuitum (two isolates) also fall under RGM and its growth looks like contamination, how author identify the same from contaminated culture?

• Author should mention the critical concentration of all drugs used in phenotypic DST. And for Moxifloxacin, WHO recommended two critical concentrations (0.25 and 1.0 ul/ml), author should also write about this.

• Abbreviate of drugs were written in very casual manner, please correct (RMP, OFL…).

• If possible, authors should also provide the data of TB-HIV co-infected patients. Because, they may have negative in sputum smear microscopy and required initial TB treatment.

• Resent molecular DST was used to identify the first line (Rifampicin and Isoniazid) drugs resistant pattern in resent study. It is advised to talk about these results in Result section also.

Reviewer #2: Clinical studies involving identification of drug resistance can be very useful in providing AMR data to the Tuberculosis Control programme of ethiopia. The results of this study has provided very less scope on scientific research outcome. However, if the author can plan for performing DNA sequencing or genotyping from the drug resistant samples. The output can be of more scientific importance and will show how drug resistant strains prevalent in that region are spreading the transmission of this disease. future work can be planned accordingly.

Reviewer #3: Ayinalem Alemu et al in their study has demonstrated a epidemiological survey among TB patients with moderate to significant resistant to 1st / 2nd line of TB Drugs.

The study is complete within its scope and I have following minor remarks to the author which need attention by the authors

1. What is underlying reason of the resistant in TB patients ? If this is clear off then this study would have global impact in term of designing suitable therapeutics for changing drug response in resistant cases.

2. Can author include some immune / blood parameters like PMN, monocyte populations, TLC / lymphocyte populations and some Th1 parameters so that we can corelate them with the resistant phenotype.

3. Ideally author should quantify the number of foamy macrophages populations which are believed to support opportunistic survival of drug resistant strain of TB and believed to have influence on the disease pattern

4. What does author mean with Poly / mono resistance among patient, how author determine this

5. whether patients resistant for 1st generation drug be resistant to 2nd generation drugs as well and vice versa ?

6. Why retreated cases are showing more resistant pattern over acute cases ?

7.from their work it is not clear why authors did not included Rapamycin / bedaquiline group in their cohort.

6. PLOS authors have the option to publish the peer review history of their article (what does this mean?). If published, this will include your full peer review and any attached files.

Reviewer #1: No

Reviewer #2: No

Reviewer #3: No

---

## [Author Response · Author response to Decision Letter 0]

28 Dec 2020

Response to Reviewers for revised manuscript number: PONE-D-20-34803

Title: The Epidemiology of first and second-line drug-resistance Mycobacterium tuberculosis complex common species: Evidence from selected TB treatment initiating centers in Ethiopia.

Revisions based on the comments and questions of all reviewers

Journal Requirements:

The revised Manuscript is corrected based on PLOS ONE style requirements. We cited all our sources including the listed references. The ORCID ID was created for the corresponding author and validated in Editorial Manager.

Reviewer 1:

While reviewing this article, actual findings are quite common and well known facts for TB prevalent countries worldwide. However, the DR-TB facts may differ for Ethiopia. Data may provide the treatment utility and recommendations in present scenario for TB Programme successfully progresses. Following are some suggestions which can make the manuscript stronger and beneficial for the readers:

Abstract:

1. Word limits excided as per Journal guidelines. Should be modified.

 Thank you for your suggestion we modified the abstract section accordingly.

2. Conclusive statement should be modified to give actual interpretation of present study. Its known fact that the higher prevalence of DR TB always associated with previously treated TB patients. 

Thank you for the advice the statement was corrected properly in the revised manuscript.

Introduction:

3. Add reference in line no. 67.

Thank you for the correction. It is a clerical error due to missing a reference, and

it is corrected in the current version.

4. Abbreviation must require for Mycobacterium tuberculosis (M. tuberculosis/ MTB) upon first appearance in the text (Line no 63).

 Thank you we abbreviated it as MTB

5. Author should advice to write few points about different molecular diagnostics modalities used presently in the referral laboratories.

Thank you we revised the manuscript based on the direction given to us.

6. Abbreviation required for XDR-TB.

 Thank you and we included the abbreviation.

Method: 

7. While recruitment of subject, authors were enrollment only TB confirmed patients using Xpert MTB/RIF and Smear microscopy, then how they get NTM among 204 TB confirmed specimens (same statement written in abstract section)? 

Thank you for the comment. In the current study, study participants were enrolled using either smear microscopy or GeneXpert. The NTM cases identified were from those patients enrolled based on the smear microscopy results that become NTM during the identification process in the culture and MPT 64 antigen test. This might be due to the fact that smear microscopy does not differentiate MTBC from NTM. 

8. Reference numbering required corrections and rearrangement. 

Thank you for the correction and we rearranged in the revised version of the manuscript.

9. Abbreviation must require for GLI, LJ, ZN, MGIT, NTM, MTBC, NTRL, BHI, MTB, PCR upon first appearance in the text.

 Thank you and corrected appropriately.

10. While writing Genotype CM, AS and MTBC assays, author should advice to write it separately. Because CM/AS kit used for NTM speciation and MTBDR V.2 assay used for identify DR-TB.

Thank you for the comment we revised the section accordingly in the Laboratory method section

11. In patient history, alcoholism and smoker status of patient as well as house hold contact should be noted if possible. Because these habits are strongly associated with TB disease and attributed to DRTB and recurrence. 

 Thank you and unfortunately we don’t have the data for the above listed variables.

Results:

12. Authors were excluded all the contaminated culture from the present study. During NTM isolation M.fortuitum (two isolates) also fall under RGM and its growth looks like contamination, how author identify the same from contaminated culture? 

For all growth detected culture tubes we inoculate in brain heart infusion agar plate Incubate brain heart infusion plate at 37ºC for 48 hours, checking for growth of contaminants at 18-24 and 48 hours. We recorded growth status in the culture reading worksheet. Besides this, for all positive culture tubes, we prepare smear stain with the Ziehl-Neelsen method. We observe the presence of Acid-fast bacilli.

If the smear is negative for AFB and the broth is clear and no growth , re-incubate the tube for further monitoring. If the AFB smear remains negative, contamination is not found on the agar plate, and the LJ slants are not positive by 8 weeks, the culture is considered to be negative. If brain heart infusion agar is contaminated and smears positive, re deconataminate by reprocessing the isolate.If brain heart infusion agar Contaminated and smear-negative the isolate was reported as contamination. Finally, if brain heart infusion agar has no growth and smear is positive for AFB, identification test performed by using MPT 64 antigen test if it is positive the organism belongs to MTBC and if negative the organism is NTM. For the detected NTM, species identification performed by using AS and CM molecular test method. 

13. Author should mention the critical concentration of all drugs used in phenotypic DST. And for Moxifloxacin, WHO recommended two critical concentrations (0.25 and 1.0 ul/ml), author should also write about this.

Thank you for the comment and we included the critical concentration of all drugs in the revised manuscript line 209-212.

14. Abbreviate of drugs were written in very casual manner, please correct (RMP, OFL…).

 Thank you for the correction. We corrected in the revised manuscript accordingly.

15. If possible, authors should also provide the data of TB-HIV co-infected patients. Because, they may have negative in sputum smear microscopy and required initial TB treatment. 

Thank you for the advice TB-HIV confection status was described in Table 1 and result section.

16. Resent molecular DST was used to identify the first line (Rifampicin and Isoniazid) drugs resistant pattern in resent study. It is advised to talk about these results in Result section also.

Thank you for valuable comment and suggestion. In this study we used molecular line probe assay technique for differentiation of species of Mycobacterium tuberculosis complex by MTBC version 2 kit and for NTM Species identification we used CM and AS assay method. For the drug susceptibility test, we used only the MGIT phenotypic DST technique.

Reviewer 2:

Clinical studies involving identification of drug resistance can be very useful in providing AMR data to the Tuberculosis Control programme of Ethiopia. The results of this study have provided very less scope on scientific research outcome. However, if the author can plan for performing DNA sequencing or genotyping from the drug resistant samples. The output can be of more scientific importance and will show how drug resistant strains prevalent in that region are spreading the transmission of this disease. future work can be planned accordingly. 

Thank you for the valuable recommendations. It is true whole-genome sequencing data provide valuable information. A large scale study should also be done on Nation Wide Drug resistance TB prevalence and sequencing was also important to identify the molecular epidemiology and phylogenic strain diversity of drug resistance TB. Meanwhile all pure positive isolates were stored in storage media at appropriate storage conditions for further analysis. For the future we will plan together with the EPHI- NTRL managers to conduct whole-genome sequencing.

Reviewer 3: 

Ayinalem Alemu et al in their study has demonstrated a epidemiological survey among TB patients with moderate to significant resistant to 1st / 2nd line of TB Drugs. The study is complete within its scope and I have following minor remarks to the author which need attention by the authors.

1. What is underlying reason of the resistant in TB patients? If this is clear off then this study would have global impact in term of designing suitable therapeutics for changing drug response in resistant cases.

Thank you for the constructive comment. There are several reasons of drug resistance in TB patients inadequate treatment, which, often the result of an irregular drug supply, prescription of inappropriate regimens, or poor treatment adherence, using as a prophylaxis especially INH, using in treatment of other bacterial disease. This is some of the factors that facilitate drug resistance and the majorities are manmade problems. 

2. Can author include some immune / blood parameters like PMN, monocyte populations, TLC / lymphocyte populations and some Th1 parameters so that we can correlate them with the resistant phenotype? 

Thank you and unfortunately we didn’t conduct immunological parameters in present study

3. Ideally author should quantify the number of foamy macrophages populations which are believed to support opportunistic survival of drug resistant strain of TB and believed to have influence on the disease pattern.

Thank you for the suggestion but we didn’t test Hematological and immunological parameters for this study.

4. What does author mean with Poly / mono resistance among patient, how author determine this

Thanks for the correction. Mono resistance TB is caused by Mycobacterium tuberculosis strains that are resistance to at least one anti TB drugs. Poly resistance TB is caused by Mycobacterium tuberculosis strains that are resistance to at least two or more anti TB drugs other than rifampicin. We define the terminology in the introduction section of the revised manuscript.

5. Whether patients resistant for 1st generation drug be resistant to 2nd generation drugs as well and vice versa?

Thanks for the comment as we described in the chart those patients resistance to first generation drugs where tested for second generation drugs those patients Sensitive for first generation drugs will not be tested for second generation drugs. We described in Table 2 and result section.

6. Why retreated cases are showing more resistant pattern over acute cases?

Thank you for the suggestion there are many factors why retreatment cases are more resistance than new cases the first reason was retreatment cases has repetitive exposure to anti TB drugs due to Relapse of case, Failure of treatment, loss to follow up and defaulting of treatment. Participant declared cured or treatment completed of any form of TB in the past, but who reports back to the health service and is now found to be AFB smear-positive or culture positive because the bacilli has a relapsing nature by escaping our immune system . Failure of treatment while on treatment, if smear-positive at the end of the fifth month or later, after commencing interruption. Return after default previously recorded as defaulted from treatment and returns to the health facility with smear-positive sputum. Transferred-in to continue treatment in a given treatment unit after starting treatment in another treatment unit for at least four weeks. e.g. Smear negative PTB case who returns after default, previously treated TB patients with an unknown outcome of that previous treatment.

7. From their work it is not clear why authors did not included Rapamycin / bedaquiline group in their cohort.

Thanks for the comment and the reason was the above listed anti TB drugs are not included in the treatment regime of drug resistance TB in Ethiopia.

---

## [Editor Report · Decision Letter 1]

6 Jan 2021

The Epidemiology of first and second-line drug-resistance Mycobacterium tuberculosis complex common species: Evidence from selected TB treatment initiating centers in Ethiopia.

PONE-D-20-34803R1

Dear Dr. Dagne,

We’re pleased to inform you that your manuscript has been judged scientifically suitable for publication and will be formally accepted for publication once it meets all outstanding technical requirements.

Kind regards,

Seyed Ehtesham Hasnain

Academic Editor

PLOS ONE

Additional Editor Comments (optional):

In this manuscript the Authors studied "The Epidemiology of first and second-line drug-resistance Mycobacterium tuberculosis complex common species: Evidence from selected TB treatment initiating centers in Ethiopia. I have gone through this revised manuscript and the Authors response to reviewers comments. There were several issues raised by the Reviewers and Authors have comprehensively addressed all the issues. Modifications have been done in the conclusive statement to give actual interpretation of the present study. Authors have included the critical concentration of all drugs in the revised manuscript line 209-212. Appropriate corrections have been made in the references. Underlying reasons for the resistance in TB patients have been clearly clarified by the Authors in the revised manuscript. All other explanations provided by the Authors to the queries of the Reviewers are quite satisfactory.
---

## [Editor Report · Acceptance letter]

14 Jan 2021

PONE-D-20-34803R1 

The Epidemiology of first and second-line drug-resistance *Mycobacterium tuberculosis* complex common species: Evidence from selected TB treatment initiating centers in Ethiopia. 

Dear Dr. Dagne:

I'm pleased to inform you that your manuscript has been deemed suitable for publication in PLOS ONE. Congratulations! Your manuscript is now with our production department. 

Kind regards, 

on behalf of

Prof Seyed Ehtesham Hasnain 

Academic Editor

PLOS ONE